

# Discovery of two skin-derived dermaseptins and design of a TAT-fusion analogue with broad-spectrum antimicrobial activity and low cytotoxicity on healthy cells

Haohao Zhu[1,2,*], Xiyan Ding[1,2,*], Wei Li[1], Tulin Lu[1], Chengbang Ma[2], Xinping Xi[2], Lei Wang[2], Mei Zhou[2], Roberta Burden[2] and Tianbao Chen[2]

[1] School of Pharmacy, Nanjing University of Chinese Medicine, Nanjing, China
[2] School of Pharmacy, The Queen's University Belfast, Belfast, United Kingdom
[*] These authors contributed equally to this work.

Corresponding authors
Wei Li, liwaii@126.com
Chengbang Ma, c.ma@qub.ac.uk

## ABSTRACT

Two novel peptides belonging to the dermaseptin family, namely DRS-CA-1 and DRS-DU-1, were encoded from cDNA libraries derived from the skin secretions of *Phyllomedusa camba* and *Callimedusa (Phyllomedusa) duellmani*. Both natural peptides are highly-conserved and exhibited high potency against wild-type Gram-positive, Gram-negative bacteria, yeast and antibiotic-resistant bacteria (MRSA and *Pseudomonas aeruginosa*) (MICs 4–8 μM) with no obvious hemolytic activity. Collectively these results suggest that both peptides may have potential as novel antibiotics. Additionally, DRS-DU-1 exhibited selective cytotoxicity to tumor cells. The truncated analogue, DP-1 and TAT-fused DP-1 (namely DP-2) were subsequently synthesised. It showed that DP-1 had low antimicrobial activity, no hemolytic and cytotoxicity to tumor cells. However, DP-2 possessed strong antimicrobial activity and the similar selective, no obvious hemolytic activity and cytotoxicity on normal human cells, but enhanced cytotoxicity to tumor cells of DRS-DU-1. These findings indicate that the N-terminus of the dermaseptins may contribute to their bioactivity, and that addition of the TAT peptide can improve biological activity. The results provide a new insight for designing novel peptide-based antimicrobial or anticancer agents with low hemolytic activity and cytotoxicity.

## INTRODUCTION

Due to the lack of efficacy of current antibiotics against increasing numbers of drug-resistant pathogens, antimicrobial peptides are considered as the compounds with the most potential to take over traditional antibiotics against drug-resistant bacteria (*Li et al., 2012*). Peptides with antimicrobial and antifungal activity have vital roles in the innate immune system, which constitutes the first line of defense against a wide range of animal-invading pathogens (*Shi et al., 2014*; *Chen et al., 2006*). The skin secretions of frogs from all around the world

have been proven to be a valuable source of these antimicrobial peptides, with more than one thousand antimicrobial peptides with significant different structural features having been extracted and characterized (*Erspamer et al., 1985*; *Azevedo-Calderon et al., 2011*).

Dermaseptins are the largest family of antimicrobial peptides identified from the skin secretions of *Phyllomedusa* species. Although they show some differences in their lengths, they are clearly related, given that almost all members are K-rich polycationic peptides, with a tryptophan residue at position 3 and a highly-conserved motif in the central or C-terminal region (*Nicolas & El Amri, 2009*). Peptides belonging to the dermaseptin family normally have two apparent separated lobes of hydrophobicity and a positively-charged electrostatic surface, resulting in the coil-to-helix transition upon association with lipid bilayers (*Chen, Tang & Shaw, 2003*; *Wang et al., 2008*). The dermaseptins usually have lytic activity and are lethal against Gram-positive and Gram-negative bacteria, fungi and protozoa at micromolar concentrations (*Shin et al., 1994*). Despite dermaseptins having identical amino acid sequences, they show significant differences in their efficiency and cytolytic activities, which was thought to be independent of the envelope structure of bacteria (*Rivas, Luque-Ortega & Andreu, 2009*; *Navon-Venezia et al., 2002*; *Huang et al., 2017*). The antimicrobial potency of dermaseptin S3 (from *Phyllomedusa sauvagii*) was not affected after shortening its chain length, the truncated N-terminal domain still retained complete or even better lytic activity against some bacteria (*Kustanovich et al., 2002*). Besides, the truncated analogues, such as $K_4$-S4(1-13)a and $K_4$-S4(1-15)a from dermaseptin S4, exhibited even lower cytotoxic effect on erythrocytes than the parent peptide (*Shepherd, Vogel & Tieleman, 2003*; *Feder, Dagan & Mor, 2000a*).

Although most dermaseptins were discovered by their antimicrobial activity, they were still reported to be effective against the cancer cells. Dermaseptin B2 and B3 (from *Phyllomedusa bicolor*) exhibited potent anti-proliferative effect on PC-3 cells, and dermaseptin B2 inhibited the tumor growth of PC-3 cells *in vivo* (*Van Zoggel et al., 2012b*; *Van Zoggel et al., 2012a*). The further study showed that dermaseptin B2 could interact with glycosaminoglycan (GAGs) on the surface of PC-3 for cell internalization (*Santos et al., 2017*). Besides, dermaseptin PH (from *Pithecopus hypochondrialis*), PD1 and PD2 (from *Pachymedusa dacnicolor*) were reported to exhibit cytotoxicity against several human cancer cell lines (*Huang et al., 2007*; *Shi et al., 2016*).

In this study, we report two novel dermaseptins, namely DRS-CA-1 and DRS-DU-1, that were discovered by molecular cloning from cDNA libraries derived from the skin secretion of *Phyllomedusa camba* and *Callimedusa (Phyllomedusa) duellmani*. The mature peptides were subsequently synthesized by solid-phase peptide synthesis and purified for functional assays. In addition, two analogues were designed based on the results to study the potential cytotoxicity to tumor cells of dermaseptins.

## MATERIALS AND METHODS

### Specimen biodata and secretion harvesting

Adult specimens of *P. camba* and *C. duellmani* were obtained from commercial sources (PeruBiotech E.I.R.L., Lima, Peru). Before harvesting the skin secretion, all frogs were

cultivated in the purpose-designed amphibian facility for at least four months, conditioned at 20−25 °C with 12h/12 h light/dark cycle and fed with multivitamin-loaded crickets three times per week. The defensive skin secretions were produced by stimulation the glands on the skin surface of frogs with gentle transdermal electrical stimulation (6V DC; 4 ms pulse-width; 50 Hz) through platinum electrodes for two periods of 20 s duration. The study was performed according to the guidelines in the UK Animal (Scientific Procedures) Act 1986, project license PPL 2694, issued by the Department of Health, Social Services and Public Safety, Northern Ireland. Procedures had been vetted by the IACUC of Queen's University Belfast, and approved on 1st March, 2011.

## Molecular cloning of the dermaseptin peptide precursor-encoding cDNAs

Five milligrams of crude lyophilized skin secretions were used to obtain polyadenylated mRNA using the Dynabeads® mRNA DIRECT™ kit (Dynal Biotech Ltd, Wirral, UK). cDNA library construction and primary cDNA amplification were performed with a BD SMART™ RACE cDNA amplification kit (BD Biosciences, UK) to give full-length prepro-peptide nucleic acid sequence data. The degenerate sense primer (S1; 5′-ACTTTCYGAWTTRYAAGMCCAAABATG-3′) (Y = C + T, W = A + T, R = A + G, M = A + C, B = T + C + G) applied in the RACE reactions were designed according to the highly conserved domain of the 5′-untranslated region of dermaseptin cDNAs from *Phyllomedusa* species. RACE products were subjected to gel analysis, and the detected bands were purified and cloned using the pGEM®-T Easy vector system (Promega, Madison, WI, USA). The sequences were obtained from an ABI 3100 automated capillary sequencer.

## Reverse phase HPLC fractionation of crude skin secretion and amino-acid sequence analysis of relevant peptides

A further five milligrams of lyophilised skin secretion were dissolved in 0.5 ml of 0.05% (v/v) trifluoroacetic acid (TFA)/water and clarified by centrifugation. The clear supernatant was subjected to reverse-phase HPLC on an analytical column (Jupiter C-5, 250 mm ×10 mm; Phenomenex, Cheshire, UK), eluted with a 0–80% linear gradient of acetonitrile containing 0.05% (v/v) TFA in 240 min at a flow rate of 1 ml/min. Absorbance was monitored at 214 nm. The peptide mapping of two natural occurring peptides was conducted using MS/MS fragmentation sequencing against the cDNA encoding peptide precursors by LCQ electrospray ion-trap mass spectrometer (Thermo Fisher Scientific, Waltham, MA, USA).

## Solid-Phase Peptide Synthesis

Both the natural peptides and the analogs were chemically synthesized through the solid-phase method with standard Rink amide resin and Fluorenylmethoxycarbonyl (Fmoc) chemistry in a PS4 automated solid-phase synthesizer (Protein Technologies, Inc, Tucson, AZ, USA). After cleavage from the resin and the protecting groups, the crude peptides were purified and analyzed by reverse-phase HPLC and MALDI-TOF mass spectrometry.

## Secondary structure and physicochemical properties prediction of the peptides

I-TASSER webserver was applied to predict the secondary structures of the synthetic peptides (*Roy, Kucukural & Zhang, 2010*). Physiochemical properties of the peptides were predicted by Heliquest and the helical wheel plots of the secondary structures were obtained from the helical wheel projections (*Gautier et al., 2008*).

In addition, circular dichroism (CD) analyses were performed on the JASCO J815 Spectropolarimeter (JASCO Inc., Easton, MD, USA). The measure range of each sample was 190–260 nm, and the parameters were set as follows: 0.5 nm data pitch, 1 nm bandwidth and 200 nm/min scanning speed. The peptide samples were prepared with 10 mM ammonium acetate buffer and 50% trifluoroethanol (TFE) in 10 mM ammonium acetate buffer at a concentration of 100 μM. The percentage of the $\alpha$-helix structure was predicted by BESTSEL online software (*Micsonai et al., 2015*).

## Antimicrobial assays

Gram-positive bacteria *Staphylococcus aureus* (NCTC10788), methicillin-resistant *Staphylococcus aureus* (MRSA) (NCTC12493), and *Enterococcus faecalis* (NCTC 12697); Gram-negative bacteria *Escherichia coli* (NCTC 10418), *Pseudomonas aeruginosa* (ATCC 27853), and *Klebsiella pneumoniae* (ATCC 43816), and the yeast *Candida albicans* (NCYC 1467) were chosen to determine the antimicrobial activities of synthesized peptides through evaluating the minimum inhibitory concentration (MIC) and minimum bactericidal concentration (MBC). The microorganisms were cultured in Muller Hinton Broth (MHB, Oxoid, UK) medium and incubated in the orbital incubator (Stuart, UK) at 37 °C overnight and then sub-cultured in a pre-warmed MHB medium until the bacteria reached their respective logarithmic growth phases. The sub-cultured suspension was diluted with fresh medium to a concentration of $5 \times 10^5$ colony forming units (CFU)/ml. The peptides with different concentrations were prepared in DMSO from 512 μM to 1 μM. The MICs were determined in 96-well microtiter plates by mixing the peptides with bacteria, and 1% DMSO in MHB was applied as the negative control. After an incubation of 20 h, the growth of microorganisms was detected at a wavelength of 550 nm in a Synergy HT plate reader (Biolise BioTek EL808, Winooski, VT, USA). The MICs were determined as the lowest concentration of peptide where no apparent growth of the microorganism was detectable. Subsequently, 10 μl of solution in each well was spotted on the Mueller Hinton Agar (MHA) plates. The MBC values were determined as the lowest concentration of peptide where no growth was observed on the MHA plate.

## Hemolysis assay

Fresh defibrinated horse blood (TCS Biosciences Ltd, Buckingham, UK) was used to perform the hemolysis assay. The serum and erythrocytes were separated by centrifugation and the erythrocytes were collected and washed with PBS. Peptides in concentration gradients were incubated with 4% (v/v) erythrocyte suspension at 37 °C for 2 h. 1% DMSO and triton X-100 (Sigma Aldrich, St. Louis, MO, USA) was used as the negative and positive control, respectively. After centrifugation, 100 μl supernatants were transferred to a 96-well

plate and absorbance measured at 550 nm using a Synergy HT plate reader (BioTec, Auburn, CA, USA). Peptide concentration causing 50% hemolysis of the red bloods cells ($HC_{50}$) was calculated [% haemolysis $= (A - A0)/ (AX - A0) \times 100$, where 'A' is absorbance with peptides of different concentrations, 'A0'is absorbance with negative controls and 'AX' is absorbance with positive controls] to determine the hemolysis activities. The therapeutic index (TI) was calculated as $HC_{50}$ divided by the geometric mean of the MIC values against relevant bacteria.

## Cell lines and cell culture

Human prostate carcinoma cell line PC-3 (ATCC-CRL-1435), non-small cell lung cancer cell line H157 (ATCC-CRL-5802), breast cancer cell line MDA-MB-435s (ATCC-HTB-129), neuronal glioblastoma cell line U251MG (ECACC-09063001) and breast cancer non-tumorigenic mammary gland cell line MCF-7 (ATCC-HTB-22) were utilized to screen the cytotoxicity of synthetic peptides. PC-3 and H157 cell lines were cultured in RPMI-1640 medium (Invitrogen, Paisley, UK), and the other cell lines were cultured with Dulbecco's Modified Eagle's medium (DMEM) (Sigma, St. Louis, MO, USA). Culture media was supplemented with 10% fetal bovine serum (FBS) (Sigma, Welwyn Garden City, UK) and 1% penicillin streptomycin solution (Sigma, UK). The human mammary epithelial cell line, HMEC-1 (ATCC-CRL-3243) was used to evaluate the cytotoxicity of the synthetic peptides against normal human cells, and cultured with MCDB131 medium (Gibco, Paisley, UK) with 10% FBS, 10ng/ml EGF, 10mM L-Glutamine and 1% penicillin streptomycin. The selected cell lines were resuscitated and transferred into a 75 cm$^2$ culture flask, and then incubated at 37 °C atmosphere containing 5% $CO_2$.

## Assessment of antiproliferative activity with the MTT assay

MTT cell viability assay was used to assess the proliferation and viability of different cell lines. After cell quantification, the cell suspension was mixed with pre-warmed medium to the final concentration of $5 \times 10^3$ cells in 96-well plates. After treating with serum-free medium, various concentrations of peptides were added and incubated for 24 h, 1% DMSO in related culture media was treated as the negative control. A total of 10 μl of MTT solution (5 mg/ml) (Sigma, Welwyn Garden City, UK) was added to each well and incubated for 4 h. Supernatants were removed by syringe and 100 μl of DMSO was used to resuspend the insoluble purple formazan crystals. The absorbance of each well was measured at 570 nm with the Synergy HT plate reader (BioTek, Winooski, VT, USA), data were analysed to obtain the mean and standard error of responses. The dose–response curves were constructed using a "best-fit" algorithm and then the half maximal inhibitory concentrations ($IC_{50}$) were calculated through the data analysis package provided in GraphPad Prism 6 (GraphPad Software, USA).

## RESULTS

### Identification of two dermaseptins from their skin secretion and bioinformatic analyses of novel peptides

Two novel peptides, namely DRS-CA-1 and DRS-DU-1, were encoded from two different cDNA libraries derived from *P. camba* and *C. duellmani*, which were the first dermaseptin

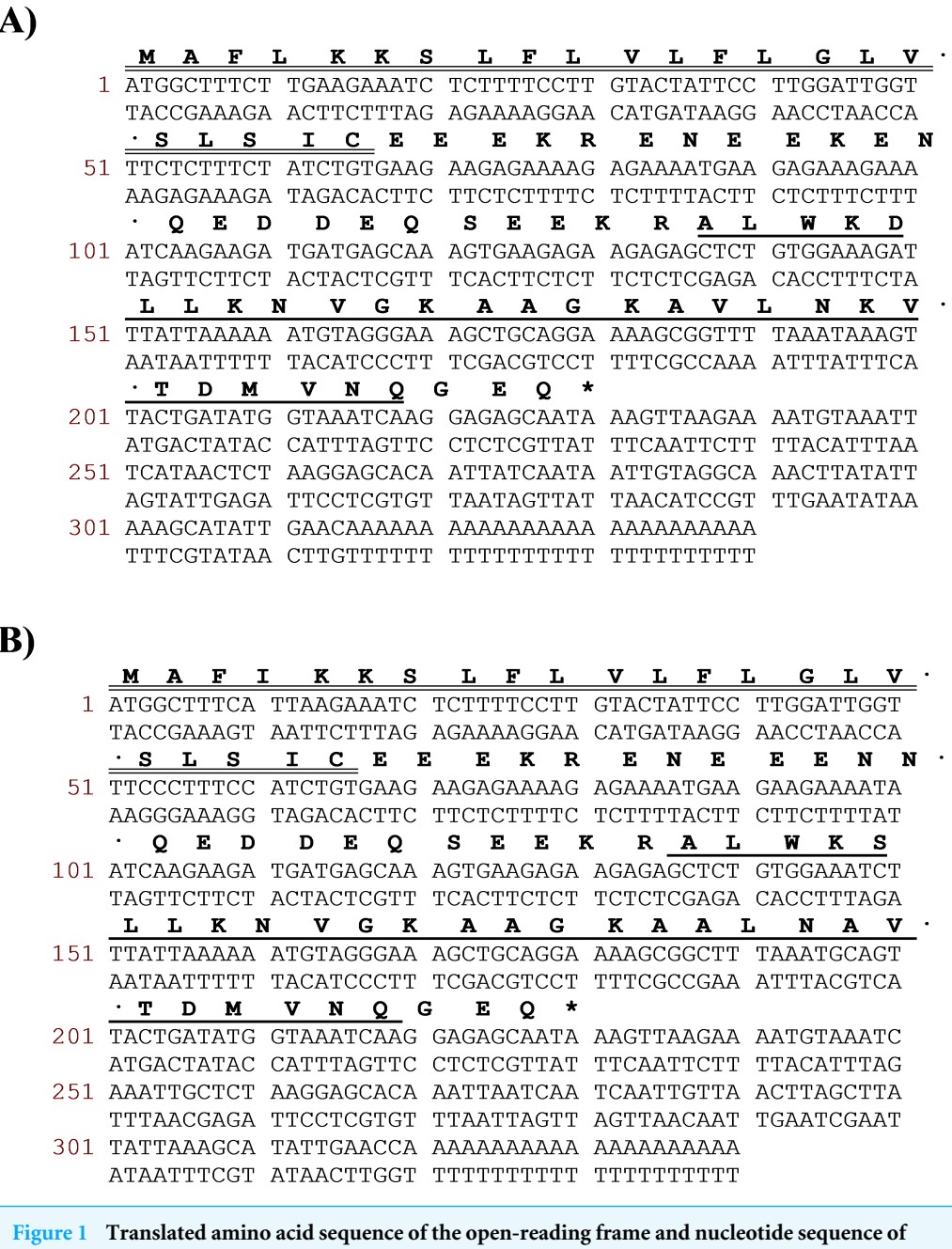

**Figure 1** Translated amino acid sequence of the open-reading frame and nucleotide sequence of cloned cDNA encoding the biosynthetic precursor of DRS-CA-1 (A) from *Callimedusa* (*Phyllomedusa*) *camba* and DRS-DU-1 (B) from *Phyllomedusa duellmani.* The putative signal peptides are marked with double line, mature peptides are marked with single line and stop codons are indicated by an asterisk.

discovered from respective frog species. The nucleotide and translated open-reading frame consisting of 76 amino acid sequences are shown in Fig. 1. Both contain 16-residue signal peptide regions, followed by a 16-residue Glu-rich acidic peptide spacer and 28-residue mature peptide with typical dermaseptin sequence, connected by two -KR- propeptide convertase processing sites. The extension (-GEQ-) was also observed, of which G acted
**Table 1  The predicted physicochemical properties and secondary structure of DRS-CA-1, DRS-DU-1, DP-1, and DP-2.** Data from physicochemical property prediction.

| Peptides | Sequence | $\alpha$-Helicity (%) | Hydrophobicity (H) | Hydrophobic moment ($\mu$H) | Net charge |
|---|---|---|---|---|---|
| DRS-CA-1 | ALWKDLLKNVGKAAGKAVLNKVTDMVNQ.NH$_2$ | 25.3 | 0.291 | 0.263 | +3 |
| DRS-DU-1 | ALWKSLLKNVGKAAGKAALNAVTDMVNQ.NH$_2$ | 27.9 | 0.331 | 0.255 | +3 |
| DP-1 | ALWKSLLKNVGKA.NH$_2$ | 16.1 | 0.429 | 0.645 | +3 |
| DP-2 | GRKKRRQRRRGALWKSLLKNVGKA.NH$_2$ | 11.4 | −0.112 | 0.358 | +11 |

as a donor for C-terminal amidation of mature peptides. LC-MS analysis confirmed the presence of DRS-CA-1 and DRS-DU-1 in the skin secretion of *P. camba* and *C. duellmani*, respectively, possessing the post-translational modification of C-terminal amidation (Figs. 2A and 2B). Alignment of the full length nucleic acid sequences and the open-reading frame sequences of the two cloned precursors are shown in Fig. 3. Both nucleotide and amino acid sequences demonstrated high degree of similarity (92%) between two biosynthetic precursors. Specifically, there was only one residue different in the signal peptides and three residues different in the mature peptides between the two moieties. A BLAST search in the Nonredundant Protein Sequence Database showed that several dermaseptins isolated from different frog species share a high degree of identity of amino acid sequences with the two dermaseptins identified herein and their open-reading frame sequences also exhibited remarkable similarities in the acidic peptides (Fig. 3). The cDNA sequences of two dermaseptin precursors have been deposited in the GenBank Database under the accession codes MF955846 and MF955847.

## Design, synthesis, physicochemical properties and secondary structure prediction of peptides and their analogues

Considering the length of dermaseptin, N-terminal of 13-mer truncated mimetic peptide, DP-1, was designed following the previous study (*Feder, Dagan & Mor, 2000b*). In the meantime, the TAT (GRKKRRQRRR) peptide was introduced at the N-terminal of DP-1 to improve its cell-penetration ability, namely DP-2. A Gly was added as a linker between TAT and DP-1. Both dermaseptins and the designed analogues were successfully synthesized and purified.

The predicted physicochemical properties and secondary structures of the peptides are shown in Table 1 and Fig. 4, respectively. Both natural peptides had 3 positive net charges and possessed $\alpha$-helical secondary structure. Additionally, the calculated $\alpha$-helicity of the natural peptides based on the CD spectra is 23.5% and 27.9%, respectively. However, DP-1 had lower degree of helicity as 16.1% though it had the same net charges and relatively higher hydrophobicity and hydrophobic moment. Whilst, DP-2 contained the lowest helical content of 11.4%. Helical wheel plots of four peptides are shown in Fig. 4C. The two natural peptides formed the hydrophobic face, which are associated with antimicrobial activity of peptides. However, both DP-1 and DP-2 revealed no uninterrupted hydrophobic face.

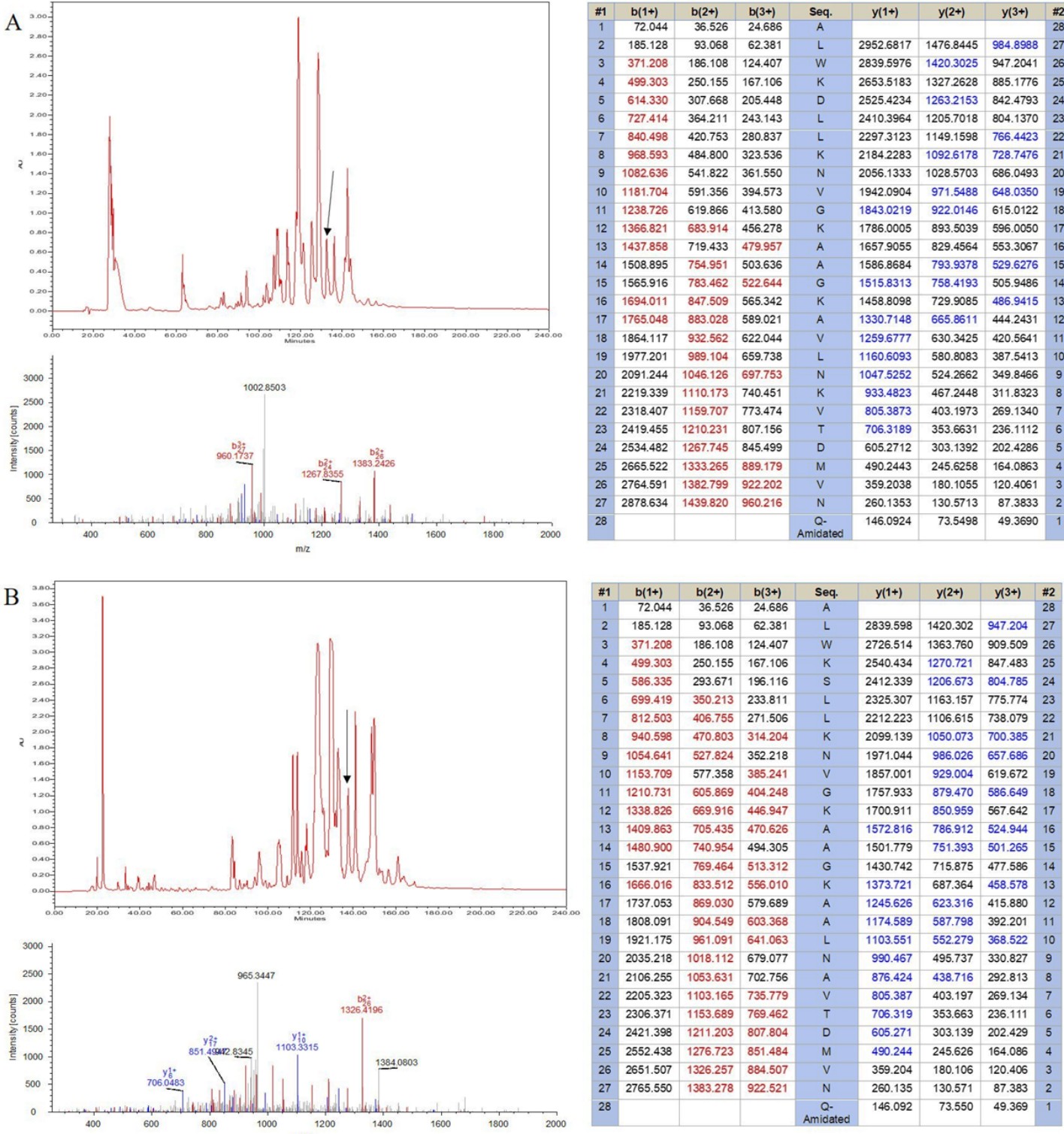

**Figure 2** **Identification of (A) DRS-CA-1 and (B) DRS-DU-1 from the corresponding skin secretions. The retention times of DRS-CA-1 and DRS-DU-1 are indicated by arrows in the respective HPLC chromatograms.** The spectra and the tables beside demonstrate the peptide mapping using the SEQUEST algorithm.

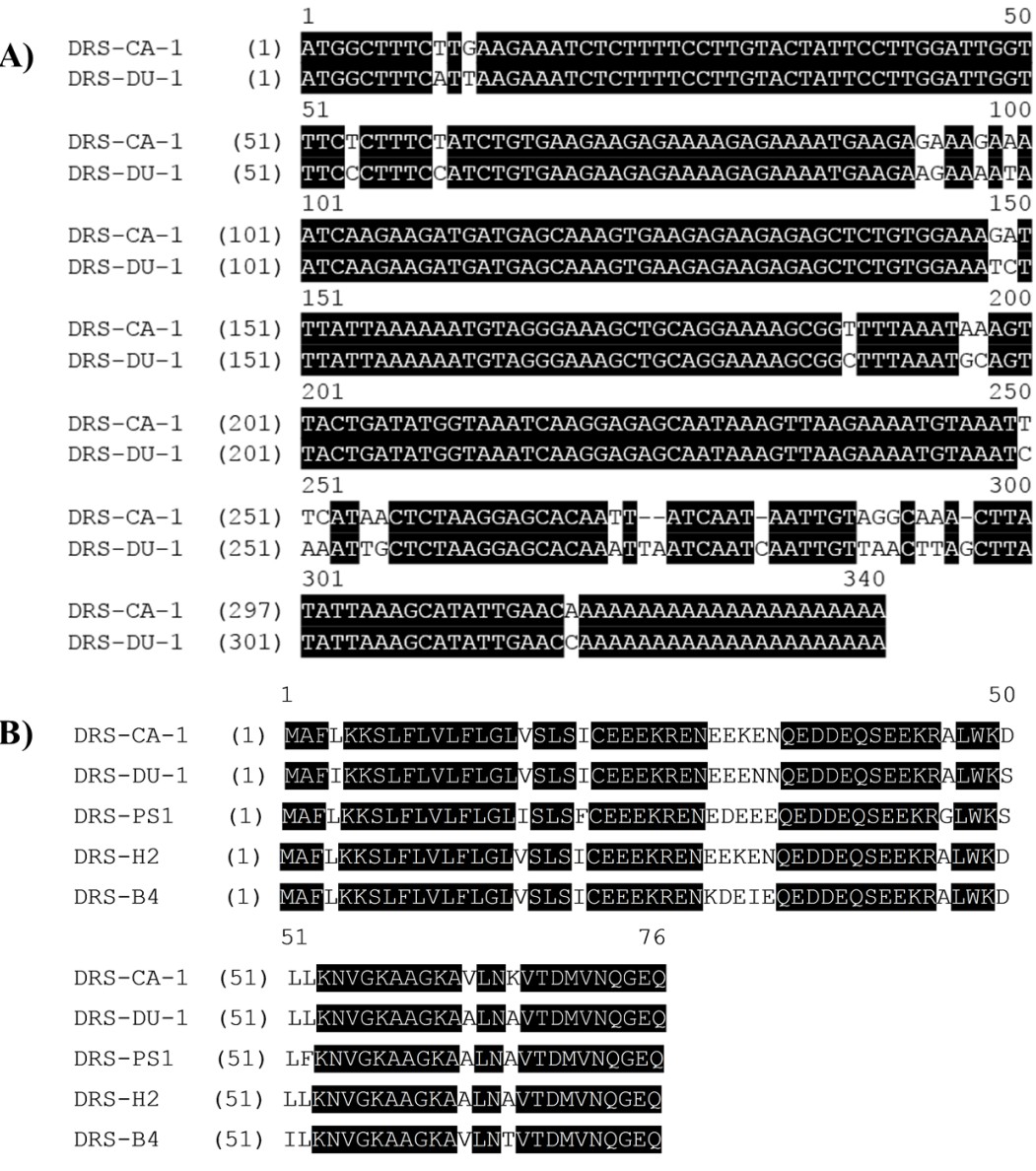

**Figure 3** **(A) Alignment of nucleotide sequences of cloned cDNAs encoding the biosynthetic precursors of DRS-CA-1 and DRS-DU-1. (B) Alignment of open-reading frame sequences of DRS-CA-1, DRS-DU-1 and other dermaseptin peptides characterized from other species.** (DRS-PS1 from *P. sauvagii* (accession number P24302), DRS-H2 from *P. hypochondrialis* (accession number P84597) and DRS-B4 from *P. bicolor* (accession number P81486)). Conserved amino acid residues are shaded in black.

## Antimicrobial and hemolytic activities

The MIC, MBC and $HC_{50}$ values of all the peptides obtained from antimicrobial and hemolysis assays are summarized in Table 2. The two natural peptides, DRS-CA-1 and DRS-DU-1, showed the same MIC values (4 μM) against *S. aureus* and *E. coli* and *C. albicans*. DRS-DU-1 was twofold more potent against MRSA, *E. faecalis*, *K. pneumoniae* and *P. aeruginosa* than DRS-CA-1. The MBC values of two natural peptides against seven tested

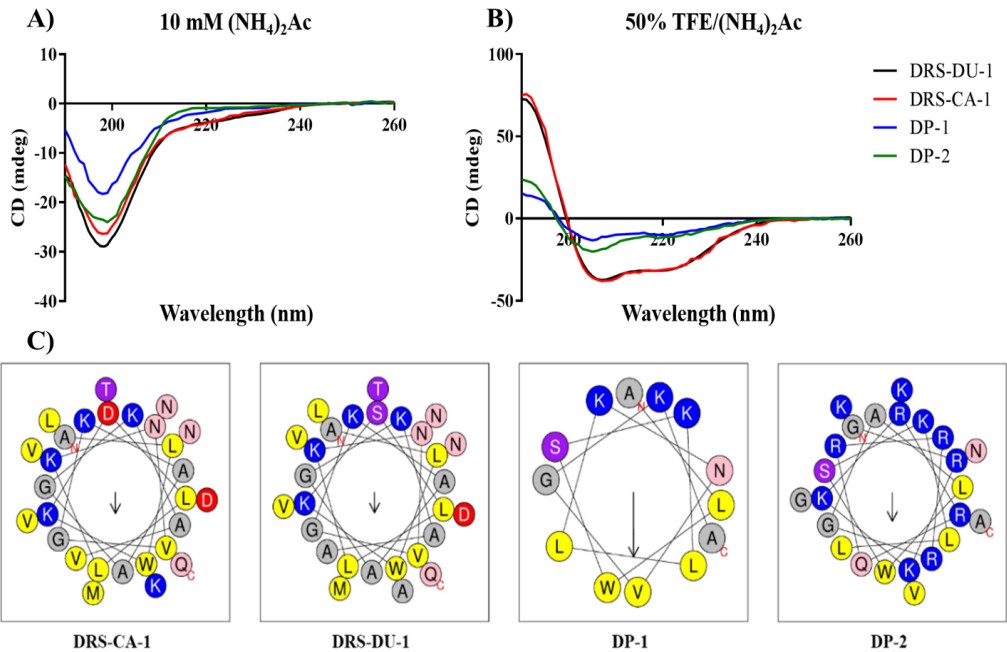

**Figure 4 CD spectra of the four peptides (100 μM) in 10 mM ammonium acetate buffer (A) and 50% TFE ammonium acetate buffer (B). Helical wheel projections (*Gautier et al., 2008*) of peptides (C), with the arrow indicated the direction of the hydrophobic moments.** All the peptides exhibit random coil structure in the aqueous solution while they are able to form $\alpha$-helical structure in the membrane-mimetic environment. The hydrophobic (yellow), hydrophilic (purple), positively-charged (blue), negatively-charged (red), amide (pink) and small (grey) residues are presented.

**Table 2 Antimicrobial (MIC and MBC) and hemolytic (HC$_{50}$) activity, and relative safety (TI) of DRS-CA-1, DRS-DU-1, DP-1, and DP-2.** Data represent the mean of ≥3 determinations.

| Strains | MIC/MBC (μM) | | | |
|---|---|---|---|---|
| | DRS-CA-1 | DRS-DU-1 | DP-1 | DP-2 |
| *S. aureus* | 4/16 | 4/16 | 64/128 | 8/8 |
| MRSA | 8/32 | 4/16 | 128/128 | 16/16 |
| *E. faecalis* | 128/256 | 64/128 | 256/>512 | 32/128 |
| *E. coli* | 4/16 | 4/16 | 64/128 | 4/8 |
| *P. aeruginosa* | 8/32 | 4/16 | 128/256 | 8/16 |
| *K. pneumoniae* | 8/128 | 4/64 | 128/>512 | 32/256 |
| *C. albicans* | 4/16 | 4/16 | 32/64 | 4/8 |
| Horse Erythrocytes (HC$_{50}$) | 114.7 | 216.6 | >512 | >512 |
| TI (overall) | 21.73 | 54.15 | 13.93 | 147.13 |

microorganisms were two or four-fold higher than respective MICs. No obvious hemolysis activity was detected at the MIC or MBC concentrations except against *E. faecalis*.

The two designed peptides also exhibited broad spectrum antimicrobial activities against the seven tested microorganisms, though the potency of DP-1 was much lower. However, the antimicrobial activities of DP-2 were restored comparing to DP-1. Additionally, the antimicrobial potency of DP-2 on *E. faecalis* was higher than the natural peptides.

**Table 3  IC$_{50}$ of DRS-CA-1, DRS-DU-1, DP-1, and DP-2 against tested human cancer cells and human normal cell.** The cytotoxicity of dermaseptin B2 (DRSB2), dermaseptin PH (DRSPH), dermaseptin PD1 (DRSPD1) and dermaseptin PD2 (DRSPD2) obtained from published data were shown in the table for comparison.

| Cell lines | IC$_{50}$ (μM) | | | | | | | |
|---|---|---|---|---|---|---|---|---|
| | DRS-CA-1 | DRS-DU-1 | DP-1 | DP-2 | DRSB2 | DRSPH | DRSPD1 | DRSPD2 |
| HMEC-1 | >100 | 53.75 | >100 | >100 | NA | 4.85 | 36.35 | 27.28 |
| H157 | >100 | 8.43 | >100 | 3.21 | NA | 2.01 | NA | 6.43 |
| PC-3 | >100 | 21.6 | >100 | 6.75 | 2.17 | 11.8 | NA | 3.17 |
| MDA-MB-435s | >100 | >100 | >100 | >100 | >10 | 9.94 | NA | NA |
| U251MG | >100 | >100 | >100 | >100 | NA | 2.36 | 15.08 | 13.43 |
| MCF-7 | >100 | >100 | >100 | >100 | NA | 0.69 | NA | NA |

**Notes.**
NA, not tested.

### Anti-Proliferative Effects of the Peptides

The anti-proliferative IC$_{50}$ values of the natural peptides and designed peptides determined by MTT assay are summarized in Table 3. DRS-CA-1 and DP-1 were found to possess no obvious anti-proliferative activity against all five human cancer cell lines and normal human cells. DRS-DU-1 had selective activity against H157 and PC-3 cells, with IC$_{50}$ values of 8.43 μM and 21.6 μM, respectively. Although DP-1 had no activity against all cancer cell lines, the activity of DP-2 was improved considerably, with the IC$_{50}$ values of 3.21 μM and 6.75 μM against H157 and PC-3 cells respectively. In addition, the cytotoxicity against normal human cell line (HMEC-1) of DRS-DU-1 (IC$_{50}$ of 53.75) was eliminated. The IC$_{50}$ values of DP-1 and DP-2 against HMEC-1 are both greater than 100 μM.

## DISCUSSION

Many bacteria have developed resistance against conventional antibiotics that has pressed researchers to discover novel antibacterial agents (*Neu, 1992*). In recent years, antimicrobial peptides isolated from amphibian skin secretions have become a hot topic in both academic and industrial drug discovery research. It is believed that most antimicrobial peptides act on the cell membrane, and can distinguish target from host cells through differences in fluidity and the negative charge density of their membranes. Such properties make it unlikely for the bacterial pathogens to become resistant to antimicrobial peptides (*Yeaman & Yount, 2003*). However, studies reported that the resistance of bacteria to AMPs has emerged recently, by producing positively-charged molecules on the membrane, and pumping AMPs out of cells (*Joo, Fu & Otto, 2016*; *Andersson, Hughes & Kubicek-Sutherland, 2016*). Therefore, discovery and design more effective antimicrobial peptide seems to be urgent and important.

In this study, we aimed to discover novel bioactive peptides from two rarely studied species of the Phyllomedusidae family, *P. camba* and *C. duellmani*. As a result, two novel dermaseptin peptides were successfully identified. The nucleotide and translated open-reading frame sequences of the peptides have almost identical signal peptide sequences,

the same prepropeptide processing site and amino donor. Furthermore, they differ by only two amino acids in the acidic peptide sequence and three residues in the mature peptides. Through the NCBI-BLAST search, it is interesting to note that the two novel precursors not only exhibit high homology to the different types of antimicrobial peptides from the Phyllomedusidae frogs, but also to other antimicrobial peptides from the Hylidae tree frogs. More than 80% identity were shared in the signal peptide domain between the two dermaseptins precursors and the antimicrobial peptides derived from the Hylidae frogs, including dermatoxin, medusin, caerin, cruzioseptin and phylloseptin (*Chen et al., 2005*; *Gao et al., 2017*; *Gao et al., 2016*; *Pukala et al., 2006*; *Proano-Bolanos et al., 2016*). This is the reason of the evolution from the ancestors that were selected by the surroundings to form diverse bioactive peptides for their unique defense system (*Vanhoye et al., 2003*).

DRS-CA-1 and DRS-DU-1 exhibited strong antimicrobial activity against Gram-positive and Gram-negative bacteria and fungi with no obvious hemolytic activity, and almost the same proportion of polar residues and nonpolar residues as well as the $\alpha$-helical structure. The positive net charge of antimicrobial peptides had been widely accepted as the key factor to help the peptide combine with the negatively-charge bacteria cell surface. Besides, the hydrophobic surface of the peptides in the $\alpha$-helical structure could ensure the peptides permeabilize the membrane. Although resistance against antimicrobial peptides have emerged, this bacterial-killing mechanism of dermaseptin is distinctly different from conventional antibiotics, which makes it promising in relation to overcoming antibiotic resistance problems (*Lee, Hall & Aguilar, 2016*). Comparing to the two most famous dermaseptins, dermaseptin B2 and S4, both natural peptides demonstrated potent antimicrobial activities, though dermaseptin B2 showed strongly inhibitory effect on *E. coli* (*Kustanovich et al., 2002*; *Joanne et al., 2009*). As the studies indicated, more cationicity of dermaseptin could improve the antimicrobial activity as well as the spectrum, which is consistent with study of introduce of Lys at positions 7 and 14 of dermaseptin S4 that remarkably increased the antimicrobial activity against *P. aeruginosa* (*Jiang et al., 2014*). Additionally, the study of truncated analogues demonstrated that the integrity of N-terminus is essential for possessing the antimicrobial activity while reducing the haemolysis within certain degree (*Kustanovich et al., 2002*). Therefore, we assumed that the 13-mer N-terminal truncated analogue, DP-1, will retain the antimicrobial potency of the patent peptides, but will become less hemolytic. Although the DP-1 had the same ratio of polar/nonpolar residues, higher hydrophobic moment and same positive net charge, it might not form the proper $\alpha$-helical structure resulting in significant reduction of its antimicrobial activity against all microorganisms. The data is different comparing to the previous study of the structure–activity relationship of dermaseptin S4 (*Kustanovich et al., 2002*), in which peptides with same physico-chemical properties usually have similar potency. This phenomenon indicates the importance of peptide's secondary structure in designing short antimicrobial peptides. Interestingly, when the TAT peptide was added, its antimicrobial activity was enhanced remarkably with reduced hemolytic activity. Considering their structural difference, this can be explained by the increased positive charges, which can enhance the interaction with the negative charge bacterial membranes. However, research has revealed that a high positive charge could lead to an increased

hemolytic activity and a loss of antimicrobial potency (*Dathe et al., 2001*), which is not corresponding with the results. We hypothesize that TAT peptide associated the attachment to the surface of bacteria, which subsequently allow the helical region of DP-1 segment to permeabilize the cell membrane.

Although dermaseptin has been reported to induce necrosis in PC-3 cancer cells, there are researches showing that dermaseptin could enter the cytoplasm and the nucleus (*Santos et al., 2017*; *Auvynet et al., 2006*). We also noticed that the sensitivity to dermaseptin varies due to different type of cancer cells. On the sensitive cells, PC-3, researches indicated that the internalization mechanism may be initiated through the interaction with negatively-charged glycosaminoglycan (GAGs) (*Santos et al., 2017*). Therefore, the disappearance of cytotoxic activity of DP-1 could be the shorter sequence and lower helicity that decreases the permeability against cell membrane, as well as the affinity to cancer cells for the GAGs mediated internalization. However, the restore of cytotoxicity activity of DP-2 suggests that fused-TAT improved the affinity of DP-2 to the cell membrane and initiated the interaction with GAGs.

On the other hand, the cytotoxicity of antimicrobial peptides on normal mammalian cells is mostly associated with the hydrophobicity and helicity of those peptides. The typical membrane lytic peptide, melittin, possesses an average helicity of 70% at membrane interfaces (*Andersson et al., 2013*). When reviewing the researches of different dermaseptins, we found that the increase of helicity not only improved antimicrobial and cytotoxic effect, but also increased the cytotoxicity on both normal cell lines and erythrocytes [12, 20]. For instance, that dermaseptin-PH showed a $\alpha$-helicity of 35% may be related to the high degree of cytotoxicity on HMEC-1 (*Huang et al., 2017*). The presence of cytotoxicity on both cancer cells and HMEC-1 of DRS-DU1 than DRS-CA1 could be also speculated by the higher $\alpha$-helicity. Additionally, researches of truncated dermaseptin analogues also indicates that the integrity of $\alpha$-helix and hydrophobic domain is important for the bio-activity of dermaseptin (*Auvynet et al., 2006*). Moreover, that the effects of DP-2 on HMEC-1 and erythrocytes were remained mild is consistent with the hypothesis of GAGs involved internalization as proportion of GAGs is larger than which is on the healthy cells.

In summary, two novel dermaseptins were deduced from the skin secretion of *P. camba* and *C. duellmani* with broad-spectrum antimicrobial activity even against drug-resistant strains (MRSA and *Pseudomonas aeruginosa*) with low hemolytic activity, which makes them promising in the treatment of multidrug-resistant bacteria. This study revealed the importance of the TAT peptide in the design of antimicrobial or anti-proliferative cancer agents through enhancing their biological function. All of these provide a new idea for designing novel peptide-based antimicrobial or anti-proliferative agents.

### Funding

Haohao Zhu and Xiyan Ding received a scholarship from the China Scholarship Council. This work was supported by the National Natural Science Foundation of China (No.

81573304), Jiangsu Collaborative Innovation Center of Chinese Medicinal Resources Industrialization (No. ZDXM-2-1) and the Priority Academic Program Development of Jiangsu Higher Education Institutions. The funders had no role in study design, data collection and analysis, decision to publish, or preparation of the manuscript.

## Grant Disclosures

The following grant information was disclosed by the authors:
National Natural Science Foundation of China: 81573304.
Jiangsu Collaborative Innovation Center of Chinese Medicinal Resources Industrialization: ZDXM-2-1.
Priority Academic Program Development of Jiangsu Higher Education Institutions.

## Competing Interests

The authors declare there are no competing interests.

## Author Contributions

- Haohao Zhu performed the experiments, analyzed the data, authored or reviewed drafts of the paper.
- Xiyan Ding performed the experiments, analyzed the data.
- Wei Li contributed reagents/materials/analysis tools, prepared figures and/or tables, approved the final draft.
- Tulin Lu contributed reagents/materials/analysis tools, prepared figures and/or tables.
- Chengbang Ma performed the experiments, analyzed the data, prepared figures and/or tables.
- Xinping Xi contributed reagents/materials/analysis tools, authored or reviewed drafts of the paper, approved the final draft.
- Lei Wang and Mei Zhou conceived and designed the experiments.
- Roberta Burden authored or reviewed drafts of the paper.
- Tianbao Chen conceived and designed the experiments, approved the final draft.

## Animal Ethics

The following information was supplied relating to ethical approvals (i.e., approving body and any reference numbers):

The study was performed according to the guidelines in the UK Animal (Scientific Procedures) Act 1986, project license PPL 2694, issued by the Department of Health, Social Services and Public Safety, Northern Ireland. Procedures had been vetted by the IACUC of Queen's University Belfast, and approved on 1st March, 2011.

## DNA Deposition

The following information was supplied regarding the deposition of DNA sequences:

The cDNA sequences of two dermaseptin precursors have been deposited in the GenBank Database under the accession codes MF955846 and MF955847.

## Data Availability

The raw data are provided in the Supplemental File.

## Supplemental Information

Supplemental information for this article can be found online at http://dx.doi.org/10.7717/peerj.5635#supplemental-information.

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
