# Peer review of "Discovery of two skin-derived dermaseptins and design of a TAT-fusion analogue with broad-spectrum antimicrobial activity and low cytotoxicity on healthy cells"

_PeerJ, doi:10.7717/peerj.5635_

## Round 0.1 · original submission · Major Revisions

I requested the opinions of three expert reviewers and thank you for your patience in awaiting their commentary. All of them seem to find that the work is worthwhile and would benefit from a number of detailed improvements, which each suggests. We look forward to reading your revised manuscript.

Reviewer 1 ·

Basic reporting

The paper is written in clear English and well-structured. Figures are of good quality.
Some typos that were found:
1. Page 1, line 30: "...system, which..."
2. Page 2, line 31: "...[2-3]. "
3. Page 3, line 82: remove "structurally determined" and write analyzed/identified. (HPLC and MS gives no indication about structure within this setup.)
4. Page 8, line 207: remove "that"
5. Page 9, line 259: insert "is" before "generally"
6. Table 2:write "Horse Erythrocytes"?

Experimental design

The experimental design is well-thought-out. Reserach question is well defined and meaningful. Methods are described with sufficient detail. However, within the caption to Figure 4, the authors should again give the reference to the software, with which they have created the helical wheel projections.

Validity of the findings

The presented data is robust and controlled.

Additional comments

I have some more remarks and comments to this work:
1. The authors should give a more detailed discussion about the possible intracellular targets of DP-2 and how they could identify them.
2. Line 194: Possibly the activity of DP-1 is much lower due to its smaller size/shorter sequence? As can be seen from the CD, DP-1 might not be able to form an adequate helix for membrane interaction?
3. Line 204: The authors should discuss why they observed reduced cytotoxicity against HUVEC when using DRS-DU1. The higher activity of DP-2 is possibly only a matter of increased positive charges?
4. Line 235. During the last years, resistancies also against AMPs have emerged. The authors should include at least one statement.

Reviewer 2 ·

Basic reporting

The article is quite clearly written but the backgound to the study is not adequately presented. There is an extensive literature on the biological properties of the dermaseptins from multiple species including antimicrobial activity and cytotoxicity to tumor cells that should be reviewed for the reader. The relevant results do not support the claim that the peptides are anticancer (see General Comments).

Experimental design

The topic of the research is within the scope of the journal and the experiments appear to have been performed to a satisfactory technical standard. A serious limitation of the study is that no attempt has been made to show that the dermaseptins are actually produced by the frogs and have the structures predicted from the nucleotide sequences of cDNAs
The frogs in question presumably synthesize a wide range of host-defense peptide including not only dermaseptins but also phylloseptins, dermatoxins, and others. The study has been submitted for publication at a too preliminary stage and would be improved enormously if a comprehensive account of the full range of peptides produced by these species were provided rather than just two peptides whose selection for publication is not explained.In its present form, the article does not fill an identified knowledge gap.
Predictions of secondary structure are not of great value. As the authors have performed CD measurements, the data should be analyzed quantitatively to determine % helicity.
.

Validity of the findings

The validity of the findings are called into question by the fact that no attempt has been made to confirm the identity of the frogs or to state where they were collected. It is simply stated that they were provided by a commercial supplier. The reader must be certain that the animals used really are the species that they are claimed to be.
In order to justify the claim that the dermaseptins are active against drug-resistant bacteria, MIC values against a wider ranger of clinical isolates whose resistance profile is fully characterized are needed.The present study involves the use of a single NCTC strain.
As regards potential as "anticancer" agents, the authors should compare the cytotoxicities against tumor cells of their peptides with those of other dermaseptins previously described.
Although the authors claim that the properties of the conjugate are "significantly better" than the native peptide, no statistical analysis is presented.

Additional comments

The authors are not justified in using the term anticancer. For an agent to be described as anticancer it must produce tumor shrinkage in humans. To use the term antitumor, the agent must produce tumor shrinkage in an animal model. The authors has shown only that the dermaseptins are cytotoxic to tumor cells in vitro.
Phllomedusa duellmani has now been reclassified as Callimedusa duellmani. and should be referred to as such.

Reviewer 3 ·

Basic reporting

The manuscript submitted by Zhu et al. describes the identification of two novel dermaseptins from two different cDNA libraries from two species of Phyllomedusa frogs. In addition, the authors report on the antimicrobial, hemolytic and anticancer properties of the natural peptides. By creating structural analogs of DRS-DU-1, the authors confirm the essential role of the N-terminal part of the peptide for the biological activity, as well as design a TAT-fusion N-terminus-truncated version with enhanced anticancer activity. The later serves as an example of strategy that can be used in designing novel antimicrobial and anticancer therapeutic agents that lack hemolytic activity.
The manuscript can be accepted for publication. However, the authors need to respond to the following comments before the paper is published:
1. I would strongly encourage the authors to think about modifying the title of the paper. The way it is written now, it is not clear and doesn’t provide enough information about the content of the paper.
2. The article overall is written in clear and unambiguous English language. However, there are some issues that need to be addressed: typos with the name of the peptides – dermaspetin (example l. 165, 233); the names of the frogs should be given as P. camba and P. duellmani throughout the text after the first time they appear in full.

Experimental design

3. The experimental design is straightforward, and the research questions are well stated.
4. Even though the information included in the Introduction is enough to justify the antimicrobial part of the work, it doesn’t familiarize the reader about the reasons for studying the anticancer effect of the peptides and their analogs on various tumor cell lines. More information needs to be included to show context.

Validity of the findings

In my opinion, the study is technically sound and the conclusions are supported by the results. It is important that the authors discuss their findings comparing them to reports by others on the antimicrobial, hemolytic and anticancer activity of dermaseptins. This peptide family is actively researched and numerous publications are available in the public domain.

Additional comments

5. In the Acknowledgements – l. 272 – 275 – the sentence is not clear.
6. It is not clear to the reviewer why the sequence in Table 1 for DRS-DU-1 is given as ALWKSLL… and for the two analogs is different, e.g. ALWSKLL… for DP-1 and TAT-GALWSKLL… for DP-2 respectively. Is it a typo? KSLL, and SKLL? As it should be the same sequence, please clarify which one is the correct one.
7. There is a typo in Table 2 – hourse ethrycytos? Needs to be corrected. In addition, it is important to add the strains of the microorganisms used as listed in the Materials and methods section, or alternatively the title of the column should be modified.
8. The authors need to define either in the Materials and methods section, or when describing the results what is therapeutic indices (TI), HC50 and IC50. The text to Table 3 – starting with “half maximal inhibitory concentrations (IC50) needs to be modidfied. It is not a proper scientific description.
9. The text to Fig. 1 contains “single-underlined” – perhaps it is a typo and needs to be modified.
10. In Fig. 2 are included sequences of other dermaseptins with their abbreviations. However, the authors must specify in the Figure text the names of the frog species these dermaseptins have been isolated from, e.g. DRS-PS1, DRS-H2, DRS-B4.
11. The text to Fig. 3 contains “membrane-mimic” – perhaps it is a typo and needs to be modified. The usage of proper terminology is encouraged.

I would be happy to review the revised manuscript.

---

## Round 0.2 · Minor Revisions

Dear Dr. Haohao Zhu,

Thank you for submitting the revised version of your manuscript. I agree with the reviewers that the manuscript is improved. There are, however, still several points and issues that have to be clarified or corrected as indicated by Reviewer 2.

We look forward to receive a re-revised version of your manuscript.
Sincerely,

Marta Kostrouchova

Reviewer 1 ·

Basic reporting

The authors worked on the paper according to comments of the reviewer. The quality of the paper highly improved. The paper is now ready for publication.

Experimental design

The authors worked on the paper according to comments of the reviewer. The quality of the paper highly improved. The paper is now ready for publication.

Validity of the findings

The authors worked on the paper according to comments of the reviewer. The quality of the paper highly improved. The paper is now ready for publication.

Additional comments

The authors worked on the paper according to comments of the reviewer. The quality of the paper highly improved. The paper is now ready for publication.

Reviewer 3 ·

Basic reporting

The quality of the revised manuscript is now better. The authors have responded to my queries. However, the text is still difficult to read at places, especially where new text/paragraphs is/are added. It is recommended that the English is checked in those places.

Experimental design

no comment

Validity of the findings

no comment

Additional comments

1. The authors had modified the text to address some of the comments from the reviewers. However, now the sentence in the abstract is confusing: "Furthermore, studies on two structure-based designed analogues, DP-1 and DP-2, showed that DP-1 had low antimicrobial activity, no hemolytic and cytotoxicity to tumor cells." Please correct the English.
2. In abstract: "These findings indicate that the N-terminal of the dermaseptins… ". It should be either N-terminus, or N-terminal part.
3. Line 50-51 - "Besides, the truncated analogue K4-S4(1-13)a and K4-S4(1-15)a demonstrated much better improvement on reducing cell lysis of erythrocytes [14, 15]." This sentence is not clear.
When referring to the dermaseptins in the Introduction, it would be useful to mention the frog species that these peptides were isolated, e.g. dermaseptin S3 (l. 48), dermaseptins B2 and B3 (l. 53), and dermaseptin PH, PD1 and PD2 (l. 57).
4. Throughout the text, the in-text references are given in a different way, e.g. [7-8] and [14,15]. This should be consistent in the whole paper. Please modify.
5. In 2.5: Secondary Structure and Physicochemical Properties Prediction of the Peptides – Please write in full first the abbreviation TFE (l. 114).
6. In 2.6: Antimicrobial assay – insert comma between Escherichia coli (NCTC 10418) Pseudomonas aeruginosa (l. 119).
l.123 – provide the source for MHB.
It is important to include a positive control for inhibition of the growth of the Gram-positive and the Gram-negative bacteria, and the yeast. Reference or a brief description for the MBC assay should be provided.
7. In 2.7: l. 133-134 "The serum and erythrocytes were separated by centrifugation and the precipitation was washed with PBS." What do you mean by precipitation? It is not the right word used here. Please correct.
8. In 2.8 l. 150: "The human mammary epithelial cell, HMEC-1" – you should add the word “line” after cell.
9. In 2.9: Assessment of Antiproliferative Activity with the MTT Assay – nothing is mentioned about positive and negative controls. In addition, more explanation of the calculation of IC50 is needed, not just “then the half maximal inhibitory concentrations (IC50) were calculated.”
10. Results:
l. 176 - “LC-MS analysis confirmed the presence of both peptides in their skin secretions” – it is not clear which skin secretions? Please modify the English.
l. 187 “the accession code, MF955846 and MF955847.” Should it be codes?
l.211 - “except which against E. faecalis.” Perhaps the word which is to be removed.
l. 214 and 215 – “Additionally, the antimicrobial potency on E. faecalis was higher than the natural peptides.” Does it apply for both designed peptide analogues?
l.223 and 224 – “In addition, the cytotoxicity against normal human cell line (HMEC-1) of DRS-DU-1 was eliminated.” It will be useful to add a value in the text as well.
11. Discussion:
l. 237 and 238 – “In this study, two species of the Phyllomedusidae family, P. camba and C. duellmani, were subjected to discover novel bioactive peptides.” This sentence is not clear – please modify.
l. 250 – “Both natural peptides exhibited strong…” – it will be useful to include the name of the natural peptides here – for clarity.
l. 266 – 268 – “Therefore, we carried out the synthesis of a 13-mer N terminal truncated analogue, DP-1, expecting the remaining of antimicrobial effect but less cytotoxicity on erythrocytes.” This sentence is not clear. I suggest that the authors modify it.
l. 282 – 304 – The whole newly added text needs attention. The English language is very heavy to read, e.g. “On the other hand, that the cytotoxicity of antimicrobial peptides on normal cells mainly depends on the hydrophobicity and helicity.”
12. References:
In [7] – the name of the frog Phyllomedusa sauvagei should be italic.
In [19] - Pithecopus (Phyllomedusa) hypochondrialis – should be italic.
In [20] - Pachymedusa dacnicolor – should be italic.
In [37] - Acinetobacter baumannii and Pseudomonas aeruginosa – the names of the bacteria should be italic.
13. Text of Table 2: I suggest that the text to Table 2 is modified to better reflect the content of the Table, e. g. "MICs, MBCs, and TI of DRS-CA-1, DRS-DU-1, DP-1, and DP-2 against tested microorganisms." to become: Antimicrobial (MIC and MBC) and hemolytic (HC50) activity, and relative safety (TI) of DRS-CA-1, DRS-DU-1, DP-1, and DP-2. In the same text, “average performance.” – this is not a scientific expression. Please modify.
14. Text to Table 3 - "were showed" - should be were shown.
15. Text to Fig. 1 - Phyllomedusa duellmeni - there is a typo - please correct it - Phyllomedusa duellmani.
16. Text to Fig. 5 - "(B)Helical wheel projections .." - insert space between (B) and helical. In the same text - negatively-charged (Red) - red should be with small letter for consistency.

---

## Round 0.3 · Minor Revisions

Dear authors,

Thank you for submitting the revised version of the manuscript. I agree with Reviewer 3 that there are still issues that have to be improved or clarified.

The main concern is the lack of negative and positive controls. These controls may include (in the case of positive controls) implementation of controls with known antibiotics with known inhibitory and bactericidal activity on the bacteria used in this study.

The problem is likely to be caused by the insufficient description of methods used for the determination of MIC, BMC as well as HC50 and IC50 and lack of both primary data and their analysis that would clearly show how the values have been obtained.

The primary data and their analysis should be submitted as Supplementary information.

Additionally, please, address all points and concerns of Reviewer 3.

Reviewer 3 ·

Basic reporting

The quality of the revised manuscript is now improved. The authors have addressed most of my comments and corrected the text accordingly. However, my main concern remains with the fact that there are no positive controls (antibiotics with known antimicrobial activity for the strains used) included in the antimicrobial assay, as well as no positive and negative controls included in the cytotoxicity studies. Despite my comments, the authors didn’t provide any clarification about those important controls.

Experimental design

no comment

Validity of the findings

no comment

Additional comments

Major concern:
1. It is important to include a positive control for inhibition of the growth of the Gram-positive and the Gram-negative bacteria, and the yeast.
2. In 2.9: Assessment of Antiproliferative Activity with the MTT Assay – nothing is mentioned about positive and negative controls. In addition, more explanation of the calculation of IC50 is needed, not just “then the half maximal inhibitory concentrations (IC50) were calculated.”

Minor comments – the paper could be published without those being modified.
3. In 3.4.: In the newly added sentence at the end of the paragraph – “The IC50 values DP-1 and DP-2 against HMEC-1 are both greater than 100μM.” add “of” after values.
4. In Discussion l. 273-275: “Therefore, we assumed that the 13-mer N-terminal truncated analogue, DP-1, might possess the similar degree antimicrobial effect to the parent peptides, but produce less cytotoxicity on erythrocytes.” I suggest the sentence to be modified as follows: “Therefore, we assumed that the 13-mer N-terminal truncated analogue, DP-1, will retain the antimicrobial potency of the parent peptides, but will become less hemolytic.”
5. Text of Table 2: hemolytic (HC50) – use subscript.
6. Text to Fig. 2: “The retention time of DRS-CA-1 and DRS-DU-1 are indicated by an arrow in the HPLC chromatogram.” Should read: ““The retention times of DRS-CA-1 and DRS-DU-1 are indicated by arrows in the respective HPLC chromatograms.”
7. Text to Fig. 5 - insert a comma after (red).

---

## Round 0.4 · Minor Revisions

Dear authors,
Thank you very much for submitting the revised version of the manuscript. There are still minor issues that need to be clarified.
Please respond to the comments of Reviewer 3 concerning the questions about the PBS used as a control in two of the experiments.

Sincerely yours,
Marta Kostrouchova

Reviewer 3 ·

Basic reporting

no comments at this point

Experimental design

It is not clear why PBS is now stated to be used as a negative control in the case of antimicrobial assays. Are the peptide dilutions made in PBS? I would have assumed that the dilutions are prepared in MHB; hence MHB should have been used as a negative control.
Once again, PBS is stated as a control for the “Assessment of Antiproliferative Activity with the MTT Assay”. Once again my question is – Why? I would have assumed that peptide dilutions are made in culture media.

Validity of the findings

no comments at this point

Additional comments

The point of using a positive control, e.g. the respective antibiotics, for inhibition of the microorganisms used in this study is NOT to compare directly the activity of the peptides to the available drugs but to demonstrate the expected behavior of the microbial strains used to respond or be resistant to the action of the antibiotics. This is rather a control that the test was set up and worked as expected, and that the microorganisms are the correct ones, e.g. sensitive or resistant.

I recommend that the manuscript is ACCEPTED for publication, after the authors respond to my questions about the PBS used as a control in two of the experiments (see queries in the section Experimental design).

---

## Round 0.5 · accepted · Accept

Thank you for revising the manuscript.

#